# Semantic-Aware and Quality-Aware Interaction Network for Blind Video Quality Assessment

## ABSTRACT

Current state-of-the-art video quality assessment (VQA) models typically integrate various perceptual features to comprehensively represent video quality degradation. These models either directly concatenate features or fuse different perceptual scores while ignoring the domain gaps between cross-aware features, thus failing to adequately learn the correlations and interactions between different perceptual features. To this end, we analyze the independent effects and information gaps of quality- and semantic-aware features on video quality. Based on an analysis of the spatial and temporal differences between two aware features, we propose a semantic-**A**ware and quality-**A**ware **I**nteraction **Net**work (A²**INet**) for blind VQA. For spatial gaps, we introduce a cross-aware guided interaction module to enhance the interaction between semantic- and quality-aware features in a local-to-global manner. Considering temporal discrepancies, we design a cross-aware temporal modeling module to further perceive temporal content variation and quality saliency information, and perceptual features are regressed into quality score by a temporal network and a temporal pooling. Extensive experiments on six benchmark VQA datasets show that our model achieves state-of-the-art performance, and ablation studies further validate the effectiveness of each module. We also present a simple video sampling strategy to balance the effectiveness and efficiency of the model. The code for the proposed method will be released.

## CCS CONCEPTS

• **Computing methodologies** → Modeling and simulation.

## KEYWORDS

Video quality assessment, semantic- and quality-aware, cross-aware guided interaction, cross-aware temporal modeling.

## 1 INTRODUCTION

The goal of video quality assessment (VQA) is to enable the model to perceive the visual quality of videos and produce results consistent with human subjective opinions, making it a popular research topic in multimedia [21, 45]. Blind VQA (BVQA) models evaluate video quality in the absence of reference videos, so huge efforts for BVQA have been devoted and a variety of deep learning-based models have been proposed [47, 59].

*MM '24, October 28–November 1, 2024, Australia, Melbourne*
© 2024 Association for Computing Machinery.
ACM ISBN 978-1-4503-XXXX-X/18/06...$15.00
https://doi.org/XXXXXXX.XXXXXXX

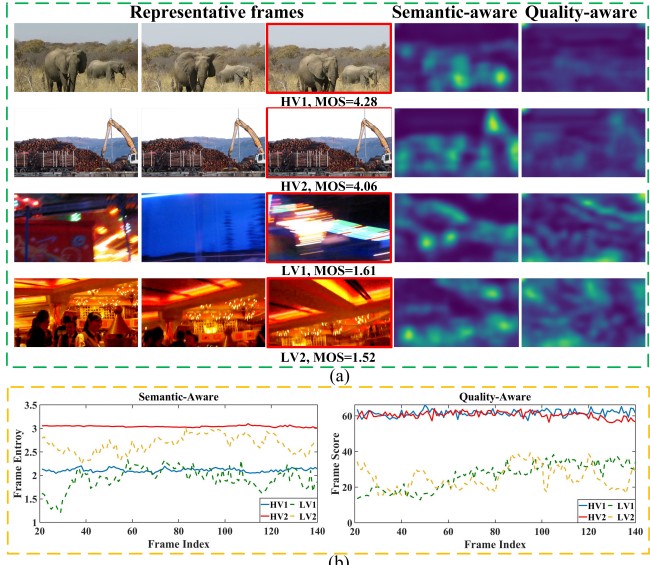

Figure 1: **Visualize the spatial feature maps of semantic- and quality-aware, alongside the temporal distribution of features on two high-quality videos (HV1 and HV2) and two low-quality videos (LV1 and LV2). MOS is the mean opinion score, higher values mean better subjective visual quality. (a) Three representative frames, and the semantic- and quality-aware feature maps are presented for the frames boxed in red. We use ResNet-50 [13] pre-trained on the ImageNet dataset [7] and the KoNIQ-10k dataset [15] to generate the semantic- and quality-aware feature maps. (b) The temporal distribution of each aware feature. Frame entropy and frame score for each frame of the feature map are used to measure information content and image quality, respectively.**

Given that human judgments of video quality are influenced by multiple perceptual factors working in concert, recent top models [47, 56] that adopt multiple networks to extract perceptual features, resulting in superior performance over models using a single network [23, 53]. These models concatenate features or fuse quality scores from different branches to comprehensively represent video quality. However, domain gaps between different aware features remain under-studied in existing work, hindering the full utilization of the advantages of multiple features in VQA and further constraining the perceptual ability of the models.

Subjective studies are instructive for the design of objective VQA models, and previous studies [43, 47] indicate that visual content and distortion artifacts play primary roles in human judgments of video quality. Visual content primarily pertains to content composition and motion information, which dominate human preferences for video content and are referred to as semantic-aware. Distortion

artifacts such as spatial degradation and temporal flicker are introduced into video due to imperfections in capture equipment or processing algorithms, which is denoted as quality-aware. Taking two sets of videos with varying quality levels, as depicted in Figure 1, as examples, we analyze the roles of semantic- and quality-aware features in VQA. **Two aware features focus on distinct spatial perceptual information by comparing videos with varying qualities from Figure 1(a).** The semantic-aware features focus on objects and are robust to quality degradation, while the quality-aware features are sensitive to quality degradation and exhibit a stronger response in low-quality frames. Thus, combining two aware features spatially can better understand the quality degradation in context. **Two aware features exhibit different temporal characteristics by observing Figure 1(b).** High-quality videos (**blue** and **red** solid curves) have slower fluctuations in temporal content information and higher frame-level quality. The opposite is true for low-quality videos (**green** and **yellow** dashed curves). Temporal content variations and overall quality emerge as key factors in discerning video quality degradation. In summary, cross-aware features exhibit perceptual gaps across spatial and temporal dimensions. We argue that simply concatenating features makes it difficult to comprehend the intricate connections between different perceptual features and video quality.

To address these challenges, we propose an effective and efficient semantic-**A**ware and quality-**A**ware **I**nteraction **Net**work ($A^2$**INet**) for BVQA. To mitigate representation differences and enhance spatial perception, we propose a cross-aware guided interaction (CAGI) module that uses cross-aware guided instance normalization (CGIN) to perceive the gaps between frame-level features with semantic- and quality-aware features, and then implements global interaction through a global self-attention layer. Motivated by Figure 1(b), we design a cross-aware temporal modeling (CATM) module by capturing information with significant quality degradation and content variation to enhance the perception for temporal distortion. Finally, the video quality score is obtained through temporal modeling and temporal pooling for the perceptual features. We conduct extensive experiments to verify that the proposed BVQA model achieves state-of-the-art (SOTA) results on the LIVE-Qual [11], CVD2014 [33], LIVE-VQC [37] and KoNViD-1k [14] datasets, with respective improvements of **7.25%, 2.43%, 4.92% and 3.95% in Spearman rank order correlation coefficient (SRCC)**. The main contributions of this paper are as follows,

1) As far as we know, this is the first attempt to explore differences in various perceptual features and employ cross-aware feature learning to enhance the perceptual ability of models for BVQA.

2) We design a CAGI module comprising a CGIN and a global self-attention layer, it uses a local-to-global manner to facilitate interaction on two aware features.

3) We introduce a CATM to perceive temporal distortions from the perspectives of quality saliency and content variation.

4) Extensive experiments have verified the advantages of the proposed BVQA model, and ablation studies have demonstrated the effectiveness of each module.

## 2 RELATED WORK

The success of the BVQA model hinges on effective perceptual feature extraction and temporal modeling to represent video quality degradation across spatial and temporal dimensions.

### 2.1 Feature extraction for BVQA

**BVQA models based on hand-crafted features.** Classic BVQA methods [6, 19, 25, 32, 35, 40] are designed to use hand-crafted features for describing visual quality degradation. Nevertheless, these hand-crafted features only emphasize low-level edges and texture information, falling short of capturing high-level semantic information. Subsequent work [20, 41] combined hand-crafted features with semantic features extracted from pre-trained ResNet-50 [13] models, yielding promising results.

**BVQA models based on deep features.** Deep learning-based models are further divided into **fixed backbone** [22, 58] and **end-to-end training** [44, 59] methods. Fixed backbone-based methods extract features from video using feature extractors pre-trained on other tasks. Li *et al.* [23, 24] used a pre-trained ResNet-50 to extract content-aware features. Related researches [3, 26] follow the work of [23, 24] using a single 2D network to capture features. Madhusudana *et al.*[31] proposed self-supervised learning to train the feature encoder. Many researches [22, 43, 51, 52, 58] have designed multiple backbone networks and concatenated features to capture various information from distorted videos, and obtained better results than a single network. Zhang *et al.* [56] incorporated five features related to visual perception and achieved SOTA performance.

For end-to-end training methods, early models [4, 53] utilized single convolutional neural networks (CNN) to extract perceptual features. Recently, Wu *et al.* [44, 45] proposed FAST-VQA, which takes video fragments as inputs and trains variants of the video swin Transformer tiny (SwinT-3D) [29] for BVQA. DisCoVQA [46] trained SwinT-3D with sparse frames as input. Yuan *et al.* [55] designed a video transformer with a multi-path temporal network and sparse attention blocks for capturing different distortions. Sun *et al.* [38] fine-tuned ResNet-50 and incorporated motion features to represent video quality. ZoomVQA [59] trained the image quality assessment (IQA) and VQA branches separately. Wu *et al.* [47] used inflated-ConvNext [28] and FAST-VQA models to perceive the quality of aesthetic and technical perspectives, obtaining video quality scores through simple weighted fusion.

Overall, existing models usually concatenate features or fuse branch scores to predict video quality. We analyze spatial gaps in different aware features to narrow the feature gaps and enhance mutual perception through a CAGI module. Since this paper focuses on exploring more effective ways of combining different aware features for BVQA rather than feature extraction, the proposed model is designed as a fixed backbone method.

### 2.2 Temporal modeling for BVQA

Temporal distortion primarily occurs in the form of flicker, jitter, and scene transitions, leading to video quality degradation. Previous studies have shown that temporal modeling is crucial for BVQA.

Li *et al.* [23, 24] modeled temporal relationships frame-by-frame using gated recurrent units (GRUs) [5], employing both min pooling and soft-weighted average pooling to aggregate frame-level

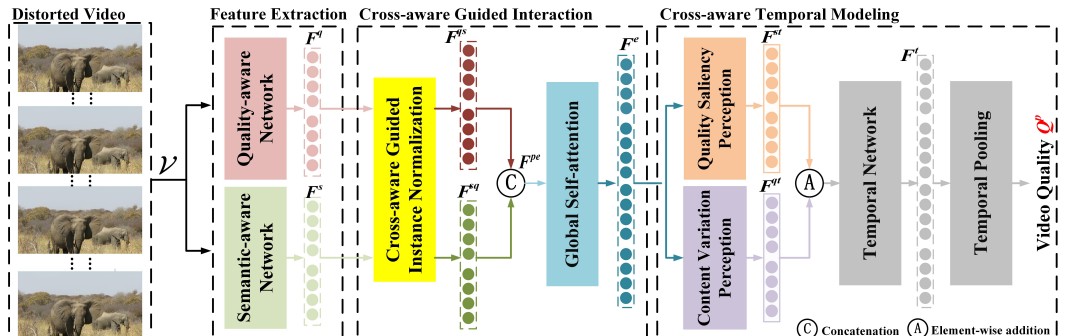

Figure 2: The framework of the proposed $A^2$INet for BVQA, with the feature extraction module (Section 3.1) to extract two aware features, the CAGI (Section 3.2) to perform local and global interactions of two aware features, and CATM module (Section 3.3) to capture the long-term dependencies of fused features and regress features into video quality scores.

scores. [22, 56] referenced this temporal modeling and temporal pooling method. Chen *et al.* [4] introduced multi-level GRUs to fuse motion information of different frequencies. Chen *et al.* [3] designed a pyramid temporal aggregation module that fuses short-term and long-term memory of frame-level features. Telili *et al.* [39] utilized bidirectional long short-term networks (Bi-LSTM) to capture temporal correlations between the previous and next frames. Ying *et al.* [52] used InceptionTime [17] for temporal modeling due to its faster and easier training. Li *et al.* [26] proposed a hierarchical Transformer that integrates frame-level and clip-level quality to derive video-level quality scores by stacking several divide and conquer Transformer layers. Wu *et al.* [46] designed a temporal content Transformer to learn the relationships among frame contents. In addition to modeling the long-term dependencies of video frames through temporal networks, some related works [22, 38, 44, 46–48, 58, 59] employed 3D networks, such as SlowFast [9], Swin-Transformer (Swin-3D) [29] and TimeSformer [1], to extract local spatio-temporal information.

In general, an effective temporal modeling module assists the model in extracting long-term dependencies and perceiving temporal distortions in videos. Existing models usually perform temporal modeling on concatenated features and do not fully exploit the gaps in representing video quality over time among different features. Based on the analysis of Figure 1(b), we propose the CATM module to perceive the temporal content variation and quality saliency information of the video, which fully considers the temporal characteristics of two aware features in videos with varying quality.

## 3 PROPOSED METHOD

Figure 2 depicts the framework of the proposed semantic-aware and quality-aware interaction network (**$A^2$INet**) for BVQA. The distorted video is first inputted into a feature extraction module to extract spatial quality-aware and semantic-aware features (Section 3.1). Subsequently, the cross-aware guided interaction (CAGI) module is employed to perceive local perceptual gaps between the two aware features and achieve global interaction (Section 3.2). Next, the fused features are fed into a cross-aware temporal modeling (CATM) module to further capture quality-aware and semantic-aware long-term dependencies, and aggregate frame-level features

through temporal pooling to estimate the video-level quality score (Section 3.3). Each module is introduced in detail below.

### 3.1 Feature Extraction

Based on previous subjective experiments regarding human preferences in visual quality [43, 47, 48], we extract spatial quality-aware features and spatial (or motion) semantic-aware features to represent video quality. The proposed model is designed a dual-branch architecture, comprising a quality-aware network and a semantic-aware network, for extracting two aware features. A distorted video $\mathcal{V} = \{\mathcal{V}(h, w, t)\} \in \mathbb{R}^{H \times W \times T}$ is treated as a collection of frame-level images, where $H \times W$ represents the spatial resolution of the video, and $T$ is the video length.

We employ an IQA model [57] pre-trained on multiple IQA datasets as the quality-aware network. Given that the visual perception system is a hierarchical structure [27, 49], features $F_t^{q1} \in \mathbb{R}^{64 \times \frac{H}{2} \times \frac{W}{2}}$, $F_t^{q2} \in \mathbb{R}^{128 \times \frac{H}{4} \times \frac{W}{4}}$, $F_t^{q3} \in \mathbb{R}^{256 \times \frac{H}{8} \times \frac{W}{8}}$ and $F_t^{q4} \in \mathbb{R}^{512 \times \frac{H}{16} \times \frac{W}{16}}$ are extracted from four bottlenecks of the quality-aware network, where $t$ represents the $t$-th frame. Then, the spatial global average pooling and global standard deviation pooling are applied for the features of each bottleneck. The pooled features are concatenated into features $f_t^q \in \mathbb{R}^{1 \times 1920}$ at the $t$-th frame, and the quality-aware features of the distorted video are represented as features $F^q = \{f_t^q\} \in \mathbb{R}^{T \times 1920}$.

For the other branch, we use the pre-trained ResNet-50 [13] on the ImageNet dataset [7] as the semantic-aware network. Similarly, we extract semantic-aware features from the distorted video, denoted as $F^s = \{f_t^s\} \in \mathbb{R}^{T \times 7680}$. To align the feature dimensions of the two aware features, we pass the features both through a fully connected (FC) layer and obtain features $F_f^s$ and $F_f^q \in \mathbb{R}^{T \times D}$.

### 3.2 Cross-aware Guided Interaction Module

We propose the CAGI, which comprises a cross-aware guided instance normalization (CGIN) and a global self-attention layer, to perceive local gaps and achieve global interaction between features.

**1) CGIN**. An adaptive instance normalization is proposed for style transfer [16] by combining the mean and variance of the features from content image $I_X$ with the features from style image

$I_Y$, which is defined as

$$\text{AdaIN}(X, Y) = \sigma(Y) \frac{X - \mu(X)}{\sigma(X)} + \mu(Y) \qquad (1)$$

where $X$ and $Y \in \mathbb{R}^{B \times C \times H \times W}$ are the features of input images $I_X$ and $I_Y$ respectively, $B$, $C$, $H$, and $W$ represent the batch size, the feature channel, the height, and width of feature map, respectively, $\mu(\cdot)$ and $\sigma(\cdot)$ calculate across spatial dimensions independently for each channel and each sample of $X$ and $Y$ [10, 30].

Inspired by this module, we use feature guided instance normalization (FGIN) to interact the cross-aware features,

$$\text{FGIN}(X, Y) = \gamma_s(Y) \frac{X - \mu(X)}{\sigma(X)} + \theta_s(Y) \qquad (2)$$

where $\gamma_s(\cdot)$ and $\theta_s(\cdot)$ denote both an FC layer, $\gamma_s(Y)$ and $\theta_s(Y)$ are treated as affine parameters to scale and shift the normalized features $X$, thereby facilitating the interaction between the two features. However, the process of deforming $X$ to $Y$ by Eq. (2) leads to the loss of the original information of $X$. For this reason, we further modify Eq. (2), as follows

$$f(X, Y) = \gamma_s(Y) \frac{X - \mu(X)}{\sigma(X)} + \theta_s(Y) + \gamma_1 \frac{X - \mu(X)}{\sigma(X)} \qquad (3)$$

where $\gamma_1$ is a constant and is set to 1. As shown in Eq. (3), we add normalized features $X$ in Eq. (3), which prevents the network from losing information $X$ during feed-forward. Finally, the proposed CGIN is formulated as follows,

$$\text{CGIN}\left(F_f^q, F_f^s\right) = f\left(F_f^q, F_f^s\right) \oplus f\left(F_f^s, F_f^q\right) = F^{qs} \oplus F^{sq} \qquad (4)$$

We narrow the perceptual gaps between the cross-aware features by deforming them towards each other, and then concatenate $F^{qs}$ and $F^{sq}$ to form the feature $F^{pe} \in \mathbb{R}^{T \times 2D}$.

**2) Global self-attention layer**. After perceiving local gaps on cross-aware features, a global self-attention layer [42, 50] is designed to enhance the global interaction between frames.

Similar to the self-attention mechanism, the feature $F^{pe}$ generates query $F_Q^{pe}$, key $F_K^{pe}$ and value $F_V^{pe} \in \mathbb{R}^{T \times 2D}$ matrices through three linear projections $W_Q$, $W_K$ and $W_V \in \mathbb{R}^{2D \times 2D}$ to encode temporal information of the $F^{pe}$,

$$F_{am}^{pe} = M_{softmax}\left(\frac{F_Q^{pe}\left(F_K^{pe}\right)^{\text{T}}}{\sqrt{2D}}\right) \in \mathbb{R}^{T \times T} \qquad (5)$$

where $M_{softmax}(\cdot)$ means a Softmax function, "T" represents the transpose operator. The values of $F_{am}^{pe}$ reflect the correlation of two elements between frames. Finally, the $F_{am}^{pe}$ is directly multiplied by the $F_V^{pe}$, and $F^{pe}$ with residual links [42] is added to obtain feature $\tilde{F}^e$.

## 3.3 Cross-aware Temporal Modeling

The CATM module is designed to capture temporal distortions associated with video quality degradation and to regress features into video-level quality scores.

Previous research has shown that frames with severe quality degradation may have a greater impact on the quality of the whole video [54]. Additionally, factors such as camera shake and scene transitions [46] also influence video quality. By integrating these

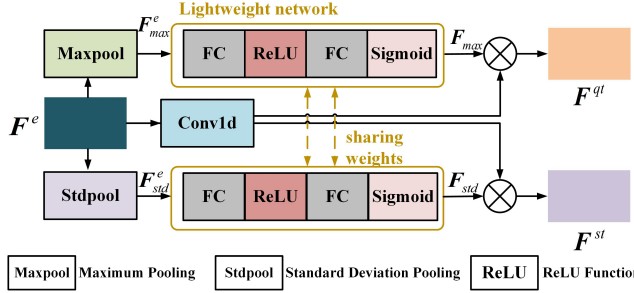

**Figure 3: Illustrations of the quality saliency perception and content variation perception blocks.**

theoretical findings with the insights from Figure 1, we develop the quality saliency perception and content variation perception blocks to enhance the perception of temporal quality, as depicted in Figure 3. Before input to the CATM module, feature $F^e$ is reduced to the $D$-dimension through an FC layer, and maximum pooling and standard deviation pooling are utilized to aggregate the saliency and variation information of feature $F^e$ along the temporal dimension. Subsequently, features $F_{max}^e$ and $F_{std}^e \in \mathbb{R}^{1 \times D}$ are fed into a lightweight network that consists of two FC layers, one ReLU function and one Sigmoid function to obtain features $F_{max}$ and $F_{std}$. Then, $F^e$ is multiplied by $F_{max}$ and $F_{std}$ after passing through a temporal convolution layer to derive the temporal quality saliency and content variation information, respectively. Finally, $F^{qt}$ and $F^{st}$ are concatenated into $F^z \in \mathbb{R}^{T \times 2D}$ and input to the temporal network to build long-term dependencies.

Similar to previous work [23], we first crop the feature $F^z$ by using a FC layer, and the reduced feature $\tilde{F}^z = \left\{ \tilde{f}_t^z \right\} \in \mathbb{R}^{T \times 128}$ is obtained. Then, the feature $\tilde{F}^z$ is input into GRUs to capture long-term dependencies. And a FC layer is used to regress the output features $H_t = \{h_t\} \in \mathbb{R}^{T \times 32}$ of GRUs into frame-level scores $q = \{q_t\} \in \mathbb{R}^{T \times 1}$. We adopt a subjectively-inspired temporal pooling strategy [36] to integrate frame-level scores $q$ to a video-level score $Q^p$.

## 3.4 Sampling Strategy and Optimization

**1) Sampling strategy**. Existing fixed backbone-based methods usually extract features for full-resolution videos, resulting in models with high complexity that increases with the resolution of the video. In this end, we propose a simple video sampling strategy based on the characteristics of semantic-aware features and quality-aware features to balance the effectiveness and efficiency of the models. On one hand, as observed from Figure 1(a), semantic-aware features focus on object and contextual semantics but has little effect on quality degradation. On the other hand, quality-aware features are sensitive to local quality degradation but lacks understanding of global semantic content. Thus, we resize each frame to preserve the original global semantics and randomly crop each frame into four 768×768 patches to ensure the original local quality. The resized frame is set to $min(H, W) = 540$ while maintaining the aspect ratio, and we strictly aligned the sample areas to ensure raw temporal variations during patch sampling [18, 44], where $min(\cdot)$

denotes the minimum operation. Finally, the imresized video and the cropped video are fed into the semantic-aware and quality-aware networks, respectively. And, the features of four patches are averaged as frame-level quality-aware features.

**2) Optimization**. During training, the proposed model is optimized using mean absolute error (MAE) loss $\mathcal{L}_M$ and rank loss $\mathcal{L}_R$ as objective functions. The MAE loss measures the distance between the predicted score $Q^p$ and the mean opinion score (MOS) $Q^m$, denoted as:

$$\mathcal{L}_M = \frac{1}{B} \sum_{i=1}^{B} \left| Q_i^p - Q_i^m \right| \qquad (6)$$

where $i$ represents the $i$-th video from the mini-batch. The differentiable rank loss function is calculated as:

$$\mathcal{L}_R = \frac{1}{B^2} \sum_{i=1}^{B} \sum_{j=1}^{B} \max\left(0, \left| \bar{Q}_{ij}^m \right| - \phi\left( Q_i^m, Q_j^m \right) \left( \bar{Q}_{ij}^p \right) \right) \qquad (7)$$

where $\bar{Q}_{ij}^s = Q_i^s - Q_j^s$, $s \in \{m, p\}$, and $\phi\left( \cdot \right)$ is defined as :

$$\phi\left( Q_i, Q_j \right) = \begin{cases} 1, & \text{if } Q_i \geq Q_j \\ -1, & \text{if } Q_i < Q_j \end{cases} \qquad (8)$$

Finally, the training loss function is represented by:

$$\mathcal{L}_{INet} = \mathcal{L}_{MAE} + \eta \mathcal{L}_{rank} \qquad (9)$$

where $\eta$ is the parameter used to balance the two losses.

# 4 EXPERIMENTS

## 4.1 Experimental Settings

**1) Implementation details.** In the experiment, the dimension parameter $D$ is set to 1024, and the balance parameter $\eta$ is set to 1. During training, we freeze the weights of the quality- and semantic-aware networks. To compare with the VQA model based on the 3D network, we replace the spatial semantic-aware network with the SlowFast [9] pre-trained on the action recognition dataset [2] as the motion semantic-aware network. Only the fast features from the last layer of the network are used as motion semantic features, with the feature dimension of $F^s$ being 512. As a distinction, we use **S+S** to represent the combination of spatial quality-aware and spatial semantic-aware, and **S+M** to represent the combination of spatial quality-aware and motion semantic-aware. Moreover, we set up two video input modes, full resolution and preprocessing, which are abbreviated as **Proposed FR** and **Proposed PR**. For **Proposed FR**, we take the whole video as the input. For **Proposed PR**, we sample 128 frames from the video at the same interval and process them as input by the preprocessing described in Section 3.4. The experiments are conducted on PyTorch [34] with single RTX 3090 GPU. The batch size is set to 8, and Adam optimizer with an initial learning rate of $2 \times 10^{-5}$ is used for training learnable parameters.

**2) Compared methods.** The performance of the proposed model is compared with sixteen VQA models, including two models combining hand-crafted and deep features (RAPIQUE [41] and CNN-VQM [20]), and fourteen deep learning-based VQA models (including eight fixed backbone models, VSFA [23], GSTVQA [3], CoINVQ [43], PVQ [52], DCVQE [26], Li *et al.* [22], CONVIQT [31] and HVS-5M [56], and six end-to-end training models, FAST-VQA [44], SimpleVQA [38], FasterVQA [45], ZoomVQA [59], DisCoVQA [46]

and VQT [55]). Note that all models are run with the source code released by the authors and are not trained with additional VQA datasets. Two RTX 3090 GPUs are used for training HVS-5M [56], FAST-VQA [44], FasterVQA [45], ZoomVQA [59] and SimpleVQA [38].

**Table 1: Summary of six benchmark VQA datasets used for experiments. These datasets cover videos with various scenes, resolutions, durations and frame rates, which can test the performance of models on various videos. "Num. Videos", "Num. Frames", "Spatial Res." and "Time Dur." mean the number of videos,the number of frames, spatial resolution and time duration, respectively.**

| Dataset | Num. Videos | Spatial Res. | Num. Frames | Time Dur. | MOS Range |
|---|---|---|---|---|---|
| CVD2014 [33] | 234 | 480p,720p | [143,830] | 10-25s | [-6.5,93.4] |
| LIVE-Qual [11] | 208 | 1080p | [358,526] | 15s | [16.6,73.6] |
| LIVE-VQC [37] | 585 | 240p-1080p | [166,1202] | 10s | [6.2,94.3] |
| YT-UGC [43] | 1142 | 360p-4k | [71,600] | 20s | [1.2,4.7] |
| KoNViD-1k [14] | 1200 | 540p | [181,240] | 8s | [1.2,4.6] |
| LSVQ [52] | 38811 | 99p-4k | [15,605] | 5-12s | [2.4,91.4] |

**3) Six benchmark VQA datasets.** Six VQA datasets with mean opinion scores (MOS) are used as benchmark datasets to test the performance of different models, including **five small-scale** datasets (CVD2014 [33], LIVE-Qual [11], LIVE-VQC [37], YT-UGC [43] and KoNViD-1k [14]) and **one large-scale** dataset (LSVQ [52]), their details are listed in Table 1. The LSVQ dataset comprises a train subset (LSVQ$_{\text{train}}$), and two test subsets (LSVQ$_{\text{test}}$ and LSVQ$_{1080p}$), containing 28056, 7182 and 3573 videos, respectively. For YT-UGC dataset, we follow [22] and 1142 videos are selected for experiments.

**4) Evaluation criteria.** For each VQA dataset, we follow the settings of [38, 44] and the videos in each dataset are partitioned into training and test sets. Specifically, we train each model on the training set and verify the performance of the model on the test set. Pearson linear correlation coefficient (PLCC) and Spearman rank-order correlation coefficient (SRCC) are used as evaluation criteria to quantify the correlation between predicted scores and subjective judgments (*i.e.*, MOS values), where SRCC reflects the monotonicity of BVQA models, and PLCC is utilized to evaluate the accuracy of BVQA models. Following [12], a nonlinear logistic function maps the predicted scores to the same scale as the MOS before calculating PLCC. The train-test splits are repeated ten times to avoid performance bias, and the median results are reported.

## 4.2 Comparison on Individual Datesets

Table 2 presents the comparison results of the proposed models and sixteen BVQA models on five small-scale datasets. In Table 2, the weighted performance of the **proposed FR(S+M)** model surpasses that of the SOTA model [56] by **2.91%** and **3.39%** in terms of PLCC and SRCC, respectively. The **proposed FR(S+M)** model outperforms prior fixed backbone methods on four small-scale datasets, achieving **7.25%**, **2.43%**, **4.92%** and **3.95%** improvements over the second-best results in terms of SRCC on the LIVE-Qual, CVD2014, LIVE-VQC, and KoNViD-1k datasets. It is worth mentioning that the **proposed FR(S+S)** model, which does not consider local spatio-temporal information, still shows competitiveness. The

**Table 2: Results on the five benchmark VQA datasets. "F" and "B" represent feature type and backbone network, "H" and "D" stand for hand-crafted and deep features, "2D" and "3D" denote 2D and 3D backbone networks. "W.A." shows the weighted-average performance over all datasets, and weights are proportional to database-sizes. The best and second best results for fixed backbone methods are highlighted in red bold and blue bold respectively. The best result for end-to-end training methods is underlined. _Italics_ indicate data sourced from original references. "-" indicates that the results are not available.**

| Type | Models | F | B | LIVE-Qual (208) | | CVD2014 (234) | | LIVE-VQC (585) | | YT-UGC (1142) | | KoNViD-1k (1200) | | W.A. (3369) | |
|---|---|---|---|---|---|---|---|---|---|---|---|---|---|---|---|
| | | | | PLCC | SRCC | PLCC | SRCC | PLCC | SRCC | PLCC | SRCC | PLCC | SRCC | PLCC | SRCC |
| Fixed-Backbone | CNN+TLVQM(MM,20) | H+D | 2D | 0.8278 | 0.8220 | 0.7795 | 0.7486 | 0.8037 | 0.7719 | 0.8225 | 0.8112 | 0.8278 | 0.8220 | 0.8185 | 0.8045 |
| | RAPIQUE(OJSP,21) | H+D | 2D | 0.7325 | 0.6927 | 0.7174 | 0.7177 | 0.7626 | 0.7879 | 0.7510 | 0.7429 | 0.8143 | 0.8060 | 0.7721 | 0.7683 |
| Fixed-Backbone | VSFA(MM,19) | D | 2D | 0.8007 | 0.7671 | 0.8735 | 0.8732 | 0.7889 | 0.7255 | 0.7737 | 0.7659 | 0.7951 | 0.7943 | 0.7926 | 0.7765 |
| | GSTVQA(TCSVT,21) | D | 2D | 0.7544 | 0.6873 | 0.8783 | 0.8718 | 0.7429 | 0.7260 | 0.7714 | 0.7758 | 0.8091 | 0.8133 | 0.7863 | 0.7817 |
| | CoINVQ(CVPR,21) | D | 2D+3D | - | - | - | - | - | - | _0.802_ | _0.816_ | _0.764_ | _0.767_ | - | - |
| | PVQ(CVPR,21) | D | 2D+3D | - | - | - | - | _0.837_ | _0.827_ | - | - | _0.791_ | _0.786_ | - | - |
| | DCVQE(ACCV,22) | D | 2D | 0.5822 | 0.7073 | 0.7601 | 0.8296 | 0.6316 | 0.7282 | 0.6599 | 0.7739 | 0.7953 | 0.8024 | 0.7054 | 0.7759 |
| | Li _et al_(TCSVT,22) | D | 2D+3D | 0.8253 | 0.8150 | 0.9043 | 0.8909 | 0.8441 | 0.8515 | 0.8237 | 0.8387 | 0.8473 | 0.8514 | 0.8413 | 0.8476 |
| | CONVIQT(TIP,23) | D | 2D | _0.802_ | _0.797_ | _0.837_ | _0.858_ | _0.817_ | _0.808_ | _0.822_ | _0.832_ | _0.849_ | _0.851_ | _0.831_ | _0.834_ |
| | HVS-5M(TCYB,23) | D | 2D+3D | 0.8218 | 0.7866 | 0.8903 | 0.8780 | 0.8470 | 0.8531 | 0.8490 | 0.8519 | 0.8536 | 0.8538 | 0.8515 | 0.8506 |
| End-to-End | FAST-VQA(ECCV,22) | D | 3D | 0.8053 | 0.8093 | 0.8727 | 0.8635 | 0.7950 | 0.7515 | 0.8102 | 0.8122 | 0.8547 | 0.8489 | 0.8275 | 0.8181 |
| | SimpleVQA(MM,22) | D | 2D+3D | 0.8304 | 0.8054 | 0.8987 | 0.8836 | 0.8329 | 0.7907 | 0.7915 | 0.7971 | 0.8521 | 0.8483 | 0.8301 | 0.8208 |
| | Faster-VQA(TPAMI,23) | D | 3D | 0.7480 | 0.7477 | 0.8564 | 0.8490 | 0.8133 | 0.7690 | 0.8135 | 0.7987 | 0.8534 | 0.8500 | 0.8266 | 0.8122 |
| | DisCoVQA(TCSVT,23) | D | 3D | _0.823_ | _0.825_ | _0.893_ | _0.897_ | _0.844_ | _0.838_ | - | - | _0.860_ | _0.863_ | - | - |
| | ZoomVQA(CVPR,23) | D | 2D+3D | 0.8222 | 0.7987 | 0.8926 | 0.8719 | 0.7677 | 0.7227 | 0.8346 | 0.8409 | 0.8280 | 0.8301 | 0.8239 | 0.8161 |
| | VQT(MM,23) | D | 3D | - | - | - | - | _0.8357_ | _0.8238_ | _0.8514_ | _0.8357_ | _0.8684_ | _0.8582_ | - | - |
| Fixed-Backbone | **Proposed PR(S+S)** | D | 2D+2D | 0.8785 | 0.8573 | 0.9182 | 0.9059 | 0.8493 | 0.8184 | 0.8472 | 0.8536 | 0.8657 | 0.8669 | 0.8610 | 0.8561 |
| | **Proposed FR(S+S)** | D | 2D+2D | 0.8764 | 0.8676 | 0.9183 | 0.9056 | 0.8484 | 0.8195 | 0.8530 | 0.8531 | 0.8668 | 0.8695 | 0.8631 | 0.8577 |
| | **Proposed PR(S+M)** | D | 2D+3D | 0.8860 | 0.8656 | 0.9128 | 0.9042 | 0.8894 | 0.8869 | 0.8408 | 0.8478 | 0.8785 | 0.8850 | 0.8705 | 0.8729 |
| | **Proposed FR(S+M)** | D | 2D+3D | 0.8865 | 0.8741 | 0.9144 | 0.9126 | 0.8952 | 0.8951 | 0.8518 | 0.8569 | 0.8812 | 0.8875 | 0.8763 | 0.8794 |

proposed **FR(S+S)** model improves the LIVE-Qual, CVD2014 and KoNViD-1k datasets by **6.45%**, **1.65%** and **1.84%** in terms of SRCC, respectively. One possible reason for the weaker performance of the proposed **FR(S+S)** model on the LIVE-VQC dataset is that local spatio-temporal information plays a significant role in this dataset. Despite freezing the feature extractors, the weighted performance of our proposed models still outperforms that of all end-to-end training methods. In summary, the proposed model achieves SOTA performance by analyzing domain gaps across aware features and enhancing the perception of two aware features in space and time through CAGI and CATM, rather than directly concatenating features. Furthermore, the weighted performance of the proposed **PR** models is only slightly lower than that of the proposed **FR** models. We will further compare their complexity in Section 4.6.

In addition to analyzing the superior performance of the proposed model, we draw several other conclusions. First, the performance of models based on 3D feature extractors is superior to that of models based on 2D feature extractors, suggesting that local spatio-temporal information is more crucial than spatial features for VQA tasks. Although models [3, 23, 26] temporally model frame-level features via temporal network, the separate extraction of spatial and temporal features fails to capture intrinsic spatio-temporal information. Second, models based on multiple feature extractors outperform those based on a single feature extractor. The utilization of multiple feature extractors enables the capture of various perceptual features, providing a richer representation for video quality degradation and yielding superior performance. Lastly, the performance of fixed backbone-based methods is comparable to that of end-to-end training methods in small-scale datasets. End-to-end training-based methods fine-tune the backbone network to extract

features related to visual quality while using only part of the video data (clips or sparse frames) as input. Fixed backbone-based methods use feature extractors pre-trained from other tasks, yet they have access to more comprehensive video data. As a result, both methods have similar prediction performance on small datasets.

## 4.3 Intra- and Cross-dataset Validation

Next, we conduct intra- and cross-dataset validation to verify the generalization ability of the proposed model. The models are trained on the $LSVQ_{train}$ subset and validated on the $LSVQ_{test}$ and $LSVQ_{1080p}$ (intra-dataset validation). For cross-dataset validation, the optimal trained model is tested on LIVE-VQC and KoNViD-1k datasets. We choose VSFA [23] GSTVQA [3], PVQ [52], Li _et al_ [22], CONVIQT [31] and HVS-5M [56] as comparison models, as they freeze the backbone network during training. Table 3 presents the comparative results. The proposed **PR(S+M)** and **FR(S+M)** models have better generalization performance. Although the proposed **FR(S+M)** model has poor cross-dataset validation performance on the LIVE-VQC dataset, it demonstrates excellent performance in both intra- and cross-dataset validation. The proposed **PR(S+S)** and **FR(S+S)** models outperform all VQA models based on the 2D backbone network, and even surpass PVQ [52], which includes a 3D feature extractor. The performance of the proposed **PR(S+S)** even outperforms the proposed **FR(S+S)** with performance by training on a large amount of video data on the $LSVQ_{test}$ dataset. Overall, the proposed models have good generalization performance.

## 4.4 Qualitative Results

To illustrate the correlation between predicted scores with subjective scores (_i.e._ MOS), we first visualize the scatter plots of the

**Table 3: Results of the intra- and cross-dataset validation on LSVQ$_{test}$, LSVQ$_{1080p}$, and two small-scale datasets. The best and second results are highlighted in red bold and blue bold.**

| Models | Intra-dataset | | | | Cross-dataset | | | |
|---|---|---|---|---|---|---|---|---|
| | LSVQ$_{test}$ | | LSVQ$_{1080p}$ | | KoNViD-1k | | LIVE-VQC | |
| | PLCC | SRCC | PLCC | SRCC | PLCC | SRCC | PLCC | SRCC |
| VSFA | 0.8050 | 0.8045 | 0.7201 | 0.6803 | 0.8174 | 0.8163 | 0.7896 | 0.7459 |
| GSTVQA | 0.7983 | 0.7985 | 0.7053 | 0.6754 | 0.7998 | 0.7954 | 0.7604 | 0.7057 |
| PVQ | *0.828* | *0.827* | *0.739* | *0.711* | *0.795* | *0.791* | *0.807* | *0.770* |
| Li *et al.* | 0.8567 | 0.8574 | 0.7825 | 0.7695 | 0.8379 | 0.8369 | 0.8136 | 0.7892 |
| CONVIQT | 0.820 | 0.821 | - | - | - | - | - | - |
| HVS-5M | 0.8702 | 0.8731 | **0.8099** | **0.7830** | 0.8445 | 0.8401 | **0.8249** | 0.8013 |
| **Proposed PR(S+S)** | 0.8564 | 0.8561 | 0.7849 | 0.7420 | 0.8446 | 0.8375 | 0.8068 | 0.7574 |
| **Proposed FR(S+S)** | 0.8539 | 0.8542 | 0.7799 | 0.7331 | 0.8341 | 0.8329 | 0.7896 | 0.7509 |
| **Proposed PR(S+M)** | **0.8805** | **0.8789** | 0.8063 | 0.7741 | **0.8598** | **0.8640** | **0.8391** | **0.8082** |
| **Proposed FR(S+M)** | **0.8779** | **0.8780** | **0.8182** | **0.7851** | **0.8640** | **0.8673** | 0.8154 | **0.8165** |

proposed **FR (S+S)** and **PR (S+M)** models on the LSVQ$_{test}$ dataset in Figure 4. We observe that most of **the scatter points** cluster around **the red line**, indicating that the scores predicted by the models are consistent with the subjective scores.

Then, we further demonstrate one successful and one failed video prediction case of the **proposed FR (S+M)** model in Figure 5. The proposed model accurately predicts video scenes with obvious semantic information but fails to predict video examples where the semantic content is not apparent. One possible explanation is that the proposed method utilizes a fixed backbone network to extract semantic features and relies on the presence of specific semantic scenarios. Overall, the quantitative results illustrate the effectiveness of the proposed model and provide inspirations for us to better handle these scenarios in the future.

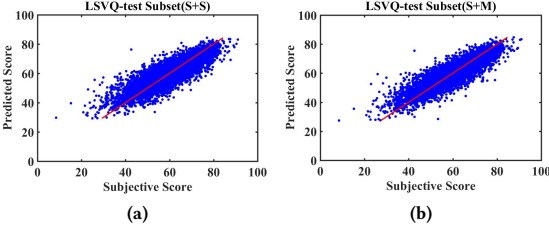

**Figure 4: Scatter plots of predicted scores on the LSVQ$_{test}$ dataset by (a) the proposed FR (S+S) model and (b) the proposed FR (S+M) model.**

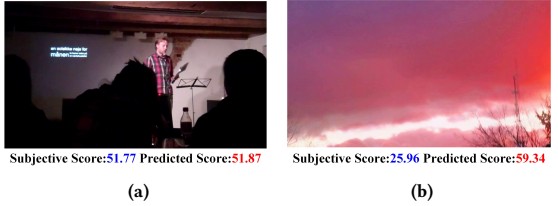

**Figure 5: (a) The one successful and (b) one failure prediction cases of the proposed FR (S+M) model.**

## 4.5 Ablation Studies

In this section, the whole video is utilized as input for ablation experiments to assess the effectiveness of each module.

**Ablation on quality-aware and semantic-aware features.** To quantitatively analyze the independent performance and mutual gains of quality-aware and semantic-aware features, we report the PLCC and SRCC results of spatial quality-aware, spatial semantic-aware, motion semantic-aware features and their combinations in Table 4. Note that all features are fed directly into a temporal network and a temporal pooling as described in Section 3.3. First, we found that the quality-aware feature typically takes a prominent role in VQA. Second, combining different aware features brings gains and improves the prediction performance of the model. Moreover, the combinations of quality-aware with semantic-aware (spatial or motion) yield superior results compared to combining two semantic-aware features, owing to the fact that they have more complementary information in distorted videos.

**Table 4: Ablation experiments on spatial quality-aware (SQ), spatial semantic-aware (SS), motion semantic-aware (MS) features and combinations (+) using three small-scale datasets. We have respectively bolded the best-performing features and combinations.**

| Features | LIVE-Qual | | LIVE-VQC | | KoNViD-1k | |
|---|---|---|---|---|---|---|
| | PLCC | SRCC | PLCC | SRCC | PLCC | SRCC |
| SQ | **0.8179** | **0.7764** | **0.8040** | **0.7599** | 0.8224 | 0.8261 |
| SS | 0.8099 | 0.7697 | 0.7918 | 0.7317 | 0.8156 | 0.8049 |
| MS | 0.6590 | 0.6159 | 0.7814 | 0.7372 | 0.7109 | 0.6993 |
| SQ+SS | 0.8193 | **0.8031** | 0.8124 | 0.7750 | 0.8390 | 0.8435 |
| SQ+MS | **0.8207** | 0.7986 | **0.8546** | **0.8526** | **0.8465** | **0.8499** |
| SS+MS | 0.8047 | 0.7728 | 0.8448 | 0.8396 | 0.8287 | 0.8278 |

**Ablation on CAGI module.** We compare three schemes that combine quality-aware features with semantic-aware features, including direct concatenation (Concat), concatenation followed by multilayer perceptron fusion (Concat+MLP), and the CAGI module. Among them, the MLP consists of an FC layer, ReLU function, and dropout layer. Then, the fused features are input into an FC layer and a temporal network as described in Section 3.3. For a full comparison, we use variants of the CNN and Transformer architectures to extract each perceptual feature and compose eight quality-semantic aware combinations for analysis. Table 5 presents the performance comparison of the three schemes. Intuitively, we observe that each feature combination through CAGI exhibits higher performance than direct concatenation and is more efficient than simple MLP fusion, proving that CAGI effectively enhances the correlation between the two aware features. Furthermore, CAGI is compatible for most dual-branch architectures.

**Ablation on CGIN.** We further investigate the impact of three instance normalizations on the proposed model, including adaptive instance normalization (AdaIN) [16], feature guided instance normalization (FGIN) [10], and the proposed CGIN. The comparison results are listed in Table 6. From Table 6, we observe that the proposed CGIN obtains higher prediction accuracy, which implies that CGIN is more helpful for the interaction of two aware features.

**Table 5: Performance comparison results using concatenation (Concat), concatenation followed by multi-layer perceptron fusion (Concat+MLP) and CAGI module combining the two aware features. We use varients of the CNN [9, 13, 57] and Transformer (ViT) [8, 29, 50] pre-trained on IQA, image classification and action recognition datasets, respectively, as backbone networks to extract spatial quality aware (CNN and ViT), spatial semantic aware (CNN and ViT), and motion semantic-aware (Fast and Swin) features. Blue and red fonts indicate quality-aware and semantic-aware features, respectively, and the performance gains of Concat+MLP and CAGI module are highlighted in green and purple.**

| Module | LIVE-VQC | | | | | | | | KoNViD-1k | | | | | | | |
|---|---|---|---|---|---|---|---|---|---|---|---|---|---|---|---|---|
| | CNN+CNN | CNN+ViT | CNN+Fast | CNN+Swin | ViT+CNN | ViT+ViT | ViT+Fast | ViT+Swin | CNN+CNN | CNN+ViT | CNN+Fast | CNN+Swin | ViT+CNN | ViT+ViT | ViT+Fast | ViT+Swin |
| Concat | 0.7750 | 0.7879 | 0.8526 | 0.7920 | 0.7824 | 0.7714 | 0.8438 | 0.7868 | 0.8435 | 0.8290 | 0.8499 | 0.8334 | 0.8405 | 0.8114 | 0.8486 | 0.8179 |
| Concat+MLP | 0.7810 | 0.8061 | 0.8698 | 0.8087 | 0.7907 | 0.7823 | 0.8536 | 0.7871 | 0.8411 | 0.8362 | 0.8694 | 0.8340 | 0.8445 | 0.8331 | 0.8582 | 0.8190 |
| Improvement | 0.77% | 2.31% | 2.01% | 2.11% | 1.06% | 1.41% | 1.16% | 0.04% | -0.28% | 0.87% | 2.29% | 0.07% | 0.48% | 2.67% | 1.13% | 0.13% |
| CAGI | 0.8159 | 0.8423 | 0.8840 | 0.8377 | 0.8264 | 0.8276 | 0.8826 | 0.8302 | 0.8665 | 0.8697 | 0.8751 | 0.8645 | 0.8584 | 0.8574 | 0.8597 | 0.8548 |
| Improvement | 5.28% | 6.90% | 3.68% | 5.77% | 5.62% | 7.29% | 4.60% | 5.52% | 2.73% | 4.91% | 2.97% | 3.73% | 2.13% | 5.67% | 1.31% | 4.51% |

**Table 6: SRCC results of different instance normalization, where "S" and "Q" mean semantic- and quality-aware features, "S→Q" represents the deformation of feature S to feature Q. The best results are bolded.**

| Module | | LIVE-Qual | | LIVE-VQC | | KoNViD-1k | |
|---|---|---|---|---|---|---|---|
| | | (S+S) | (S+M) | (S+S) | (S+M) | (S+S) | (S+M) |
| AdaIN | S→Q | 0.8513 | 0.7915 | 0.7824 | 0.8774 | 0.8514 | 0.8658 |
| | Q→S | 0.8442 | 0.8508 | 0.7642 | 0.8727 | 0.8537 | 0.8715 |
| FGIN | S→Q | 0.8485 | 0.8611 | 0.7914 | 0.8820 | 0.8583 | 0.8750 |
| | Q→S | 0.8398 | 0.8658 | 0.7897 | 0.8794 | 0.8587 | 0.8776 |
| CGIN | | **0.8676** | **0.8741** | **0.8195** | **0.8951** | **0.8695** | **0.8875** |

**Table 7: Ablation on each component of the proposed model using three VQA datasets. We choose the VSFA [23] model as our baseline. Each component improves the performance of the model. The best results are bolded.**

| Model | F E | G C | T M | LIVE-Qual | | LIVE-VQC | | KoNViD-1k | |
|---|---|---|---|---|---|---|---|---|---|
| | | | | PLCC | SRCC | PLCC | SRCC | PLCC | SRCC |
| Baseline | ✓ | ✗ | ✗ | 0.8007 | 0.7671 | 0.7889 | 0.7255 | 0.7951 | 0.7943 |
| Proposed (S+S) | ✓ | ✗ | ✗ | 0.8193 | 0.8031 | 0.8124 | 0.7750 | 0.8390 | 0.8435 |
| | ✓ | ✓ | ✗ | 0.8657 | 0.8587 | 0.8390 | 0.8159 | 0.8628 | 0.8665 |
| | ✓ | ✓ | ✓ | **0.8764** | **0.8676** | **0.8484** | **0.8195** | **0.8668** | **0.8695** |
| Proposed (S+M) | ✓ | ✗ | ✗ | 0.8207 | 0.7986 | 0.8546 | 0.8526 | 0.8465 | 0.8499 |
| | ✓ | ✓ | ✗ | 0.8785 | 0.8653 | 0.8905 | 0.8840 | 0.8711 | 0.8751 |
| | ✓ | ✓ | ✓ | **0.8865** | **0.8741** | **0.8952** | **0.8951** | **0.8812** | **0.8875** |

**Ablation on each component of the proposed model.** We investigate the effectiveness of each module, including the feature extraction module (FE), the CAGI module (GI), and the CATM module (TM). Table 7 lists the performance results of different model designing on three datasets. Based on the qualitative and quantitative analysis of the two aware features, we meticulously designed each module, and Table 7 proves that each module has certain gains for the proposed model.

## 4.6 Computational Complexity

In this subsection, we compare the FLOPs and running times on GPU of the proposed models with existing models, and plot the performance curves of FLOPs and running times on videos with different resolutions in Figure 6. Average results of ten video samples (10 seconds, 30 frames per second) as final test time, and experiments are executed on a single 3090 GPU. It can observe that the FLOPs and running time of the existing models increase with the

rise in video resolution. Although HVS-5M [56] exhibits good prediction performance, the computational complexity is unacceptable. When video resolution exceeded 540P, the HVS-5M run for more than 100s and 2160P video could not be tested on a single 3090 GPU (540P, 134.54s; 720P, 224.08s; 1080P, 388.82s; 1440P, 687.7s). The **proposed FR(S+S)** and **FR(S+M)** models achieve SOTA performance with computational complexity between VSFA [23] and GSTVQA [3]. The **proposed PR(S+S)** and **PR(S+M)** models reduce redundant information from the input video through a simple sampling strategy (as described in Section 3.4), achieving the lowest computational complexity while maintaining performance with the whole video as input (see Table 2).

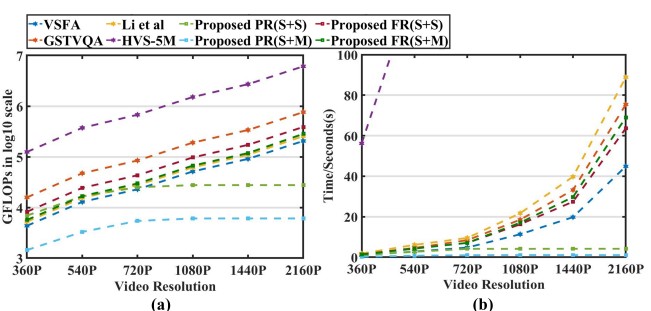

**Figure 6: (a) FLOPs and (b) running time performance curves of the proposed models and four models on videos with different resolutions.**

## 5 CONCLUSION

In this paper, we have presented the semantic-**A**ware and quality-**A**ware **I**nteraction **Net**work (**A$^2$INet**) for blind video quality assessment (BVQA). Based on the analysis of the domain gaps between semantic- and quality-aware features, we design a cross-aware guided interaction module to enhance the interaction between the two aware features and propose a cross-aware temporal modeling module to perceive temporal distortion from the perspectives of content variation and quality saliency. Experimental results show that the proposed model outperforms the state-of-the-art performance on six benchmark VQA datasets, and ablation studies further verify the effectiveness of each component. Additionally, we demonstrate a simple video sampling strategy to balance the effectiveness and efficiency of the proposed model.

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
