# OpenReview forum: "Semantic-Aware and Quality-Aware Interaction Network for Blind Video Quality Assessment"
_acmmm.org/ACMMM/2024/Conference — MM2024 Poster_

### Official Review · Reviewer_fRRQ · 2024-04-28

**Rating:** 4
**Confidence:** 3

**Summary:**

By analyzing the independent impact and information gap of quality and semantic perception features on video quality, this paper proposes a semantic perception and quality perception interaction network for blind video quality assessment (BVQA).

**Strengths:**

The method proposed in this paper is relatively novel and technically correct.

**Limitations:**

1. The paper has formatting errors. For example, the Tables and Figures should be placed at the top.
2. Please demonstrate the prediction performance of the method that combines spatial quality aware, spatial semantic aware, and motion semantic aware.
3. What are the preprocessing operations mentioned in the implementation details? For the sake of fairness, the preprocessing of the input of the comparison method should also be consistent with the preprocessing of the input of the proposed method.
4. Add the experimental results of the end-to-end comparison methods in Table 3.
5. Add ablation experiments on the loss function.

**Suitability:**

3

---

### Official Review · Reviewer_P1Fo · 2024-05-19

**Rating:** 4
**Confidence:** 3

**Summary:**

This paper explored the interaction between semantic awareness and quality awareness, and proposed a new BVQA model that combines the advantages of both types of features.

**Strengths:**

This paper effectively integrates semantic-aware and quality-aware features, addressing the common issue of simply concatenating them. The experiments are ample and rigorous, validating the proposed model's effectiveness. Most importantly, it reduces computational complexity.

**Limitations:**

1. Line 130-133 “High-quality videos (blue and red solid curves) have slower fluctuations in temporal content information ...... The opposite is true for low-quality video”. This poses a problem. As illustrated in Fig.1, there is a high frame-to-frame similarity in the two high-quality videos, leading to minimal entropy fluctuations. In contrast, the two low-quality videos show considerable motion, resulting in more pronounced entropy fluctuations. These fluctuations seem more tied to the video content rather the video quality. While I concur with the conclusion that temporal content variations and overall quality are critical in identifying video quality degradation (line 134), the provided examples are inappropriate.  In fact, using the videos in supplementary material Fig.3, the HV1 and the LV3 better support the conclusion.
2. The paper frequently mentions "narrowing representation differences." (line 142, 218, 377 etc.). However, semantic-aware and quality-aware features are inherently different by nature and are supposed to be different. What we need to do is combine the different advantages of these two types of features (which is indeed what the authors have done), rather than narrowing their differences. Please consider revising the wording.
3. In Fig. 1, Fst and Fqt are reversed. Additionally, the figure shows Fst and Fqt being added element-wise, whereas the text on line 435 mentions “concatenation”, which is inconsistent. Besides, it is recommended to place the overall output Q of the model outside the module.
4. The blue-green blocks in Fig. 3 is not labeled.
5. The full name of the model is needed only once in the abstract and the body of the text. Line 281and line 916 are redundant.
6. Fig.4, as well as the Fig.3 in supplementary material could be better with scatterplots fit lines’ slopes and intercepts, to illustrate the performance differences across datasets.

**Suitability:**

2

---

### Official Review · Reviewer_25qz · 2024-05-24

**Rating:** 2
**Confidence:** 2

**Summary:**

The paper introduces A2INet, a semantic-Aware and quality-Aware Interaction Network for blind video quality assessment (BVQA). It addresses the domain gaps between semantic and quality features by using a cross-aware guided interaction module and a cross-aware temporal modeling module. A2INet enhances feature interaction and models temporal distortions to improve BVQA.

**Strengths:**

1. The logic is clearly expressed, facilitating understanding.

2. The experimental results indeed demonstrate some of the article's advantages.

**Limitations:**

1. I am concerned about the role of semantic features, as this might influence the genuine evaluation of quality features; I think we should assess the various roles played by quality information and semantic information before passing through the Cross-aware Guided Instance Normalization module.  If there is any misunderstanding, I welcome the authors to refute it.

2. Does the module's outcome solely focus on global features while neglecting local ones?

3. The majority of the formulas are missing punctuation.

4. Wherein lie the advantages of using semantic features in conjunction with quality-aware features, as opposed to interacting with global and local features?

5. What are the advantages of using a cross-perception guided interaction module, as claimed by the authors, compared to this method[1] of training the interaction between semantic-aware and quality-aware features using contrastive learning?

6. Please analyze the computational complexity and the resulting computational cost of the proposed method.

[1] Zhao K, Yuan K, Sun M, et al. Quality-aware pre-trained models for blind image quality assessment[C]//Proceedings of the IEEE/CVF Conference on Computer Vision and Pattern Recognition. 2023: 22302-22313.

**Suitability:**

2

---

### Meta-Review · Area_Chair_5fwp · 2024-07-08

**Recommendation:** Accept (Poster)
**Confidence:** 3

**Metareview:**

All reviewers appreciated clarifications in the authors' rebuttal, and the paper is now more toward "borderline accept". The main remaining concern is about the novelty of the approach leading to "borderline accept".